# Meta-Analysis and Validation of a Colorectal Cancer Risk Prediction Model Using Deep Sequenced Fecal Metagenomes

**DOI:** 10.3390/cancers14174214

**Published:** 2022-08-30

**Authors:** Mireia Obón-Santacana, Joan Mas-Lloret, David Bars-Cortina, Lourdes Criado-Mesas, Robert Carreras-Torres, Anna Díez-Villanueva, Ferran Moratalla-Navarro, Elisabet Guinó, Gemma Ibáñez-Sanz, Lorena Rodríguez-Alonso, Núria Mulet-Margalef, Alfredo Mata, Ana García-Rodríguez, Eric J. Duell, Ville Nikolai Pimenoff, Victor Moreno

**Affiliations:** 1Unit of Biomarkers and Suceptibility (UBS), Oncology Data Analytics Program (ODAP), Catalan Institute of Oncology (ICO), L’Hospitalet del Llobregat, 08908 Barcelona, Spain; 2ONCOBELL Program, Bellvitge Biomedical Research Institute (IDIBELL), L’Hospitalet de Llobregat, 08908 Barcelona, Spain; 3Consortium for Biomedical Research in Epidemiology and Public Health (CIBERESP), 28029 Madrid, Spain; 4Digestive Diseases and Microbiota Group, Girona Biomedical Research Institute (IDIBGI), Salt, 17190 Girona, Spain; 5Department of Clinical Sciences, Faculty of Medicine, University of Barcelona, 08007 Barcelona, Spain; 6Gastroenterology Department, Bellvitge University Hospital, L’Hospitalet de Llobregat, 08907 Barcelona, Spain; 7Medical Oncology Department, Catalan Institute of Oncology (ICO), 08916 Badalona, Spain; 8Badalona-Applied Research Group in Oncology, Catalan Institute of Oncology (ICO), 08916 Badalona, Spain; 9Digestive System Service, Moisés Broggi Hospital, 08970 Sant Joan Despí, Spain; 10Endoscopy Unit, Digestive System Service, Viladecans Hospital-IDIBELL, 08840 Viladecans, Spain; 11Department of Laboratory Medicine, Karolinska Institutet, 14186 Stockholm, Sweden

**Keywords:** colorectal cancer, microbiome, shotgun, metagenomics, MWAS, meta-analysis, predictive model

## Abstract

**Simple Summary:**

Colorectal cancer (CRC) is the third most common cancer in the world. The gut microbiome, which includes a collection of microbes, is a potential modifiable risk factor. The study of the microbiome is complex and many issues remain unsolved despite the scientific efforts that have been recently made. The present study aimed to build a CRC predictive model performing a meta-analyses of previously published shotgun metagenomics data, and to validate it in a new study. For that purpose, 156 participants of a CRC screening program were recruited, with an even distribution of CRCs, high-risk colonic precancerous lesions, and a control group with normal colonic mucosa. We have identified a signature of 32 bacterial species that have a good predictive accuracy to identify CRC but not precancerous lesions. This suggests that the identified microbes that were enriched or depleted in CRC are merely a consequence of the tumor.

**Abstract:**

The gut microbiome is a potential modifiable risk factor for colorectal cancer (CRC). We re-analyzed all eight previously published stool sequencing data and conducted an MWAS meta-analysis. We used cross-validated LASSO predictive models to identify a microbiome signature for predicting the risk of CRC and precancerous lesions. These models were validated in a new study, Colorectal Cancer Screening (COLSCREEN), including 156 participants that were recruited in a CRC screening context. The MWAS meta-analysis identified 95 bacterial species that were statistically significantly associated with CRC (FDR < 0.05). The LASSO CRC predictive model obtained an area under the receiver operating characteristic curve (aROC) of 0.81 (95%CI: 0.78–0.83) and the validation in the COLSCREEN dataset was 0.75 (95%CI: 0.66–0.84). This model selected a total of 32 species. The aROC of this CRC-trained model to predict precancerous lesions was 0.52 (95%CI: 0.41–0.63). We have identified a signature of 32 bacterial species that have a good predictive accuracy to identify CRC but not precancerous lesions, suggesting that the identified microbes that were enriched or depleted in CRC are merely a consequence of the tumor. Further studies should focus on CRC as well as precancerous lesions with the intent to implement a microbiome signature in CRC screening programs.

## 1. Introduction

Colorectal cancer (CRC) is the third most common cancer in the world, and the second with the highest mortality [1]. Around 20% of CRC patients have family history or inherited syndromes that predispose them to CRC, such as Lynch Syndrome or Familial Adenomatous Polyposis. However, the vast majority of CRCs are considered sporadic, arising in the context of sequential multistep genomic aberrations and the influence of modifiable risk factors [2]. The gut microbiome is considered to be one of such factors, as gut epithelial microbes have an important role in host intestinal homeostasis [3]. Although the specific mechanisms through which the gut microbiota affect the host intestinal metabolism are still a matter of study, gut microbes are known to interact with both inflammatory and metabolic functions of the host in ways that are relevant to CRC development [4,5]. For instance, dysbiosis is thought to facilitate the growth of pathogenic species, inflammation, and alteration of the immune system with effects on cell proliferation [6,7]. In particular, several studies have observed positive associations with CRC risk in presence of *Fusobacterium nucleatum, Bacteroides fragilis,* and *Escherichia coli*, among others [8].

Sequencing of the 16S rRNA gene has been long used to investigate the gut microbiome. However, whole shotgun metagenomic sequencing is an alternative that allows the achieving of systematic species-level resolution and provides information from the whole bacterial genome, which are limitations of the former [9]. While a large part of the human gut microbiome remains unknown [10], recent bioinformatics efforts have allowed the assembly of hundreds of genomes from shotgun metagenomics data. These metagenome-assembled genomes (MAGs) expand our knowledge of the gut microbiome [11,12,13]. MAGs have also allowed researchers to create new databases, such as the genome taxonomy database (GTDB), which creates a new phylogeny-based bacterial and archaeal taxonomy [14], and the Unified Human Gastrointestinal Genome (UHGG), a database containing exclusively fecal microbes, including MAGs [15].

In regards to CRC, several case-control studies [16,17,18,19,20,21], including three meta-analyses [22,23,24] have explored microbial signatures that are relevant to CRC by performing microbiome-wide association studies (MWAS), where each bacterium or archaea that was detected in samples was tested for association with CRC. Bacteria that were found to be related to CRC in these studies included *Porphyromonas asaccharolytica*, *Parvimonas micra*, *Peptostreptococcus stomatis*, *Gemella morbillorum,* and *Solobacterium morei*, among others. Beyond getting knowledge on which bacteria are related to CRC, the results of MWAS can be used to build risk prediction models with an interest in improving CRC screening programs; however, further evidence is needed before its implementation [25].

The present study served two aims. First, to build a CRC predictive model as result of the meta-analysis of the previously published shotgun metagenomics data. Second, to validate this model in a new study with 156 participants that were recruited in a CRC screening context, selected with an even distribution of CRCs, high-risk precancerous colonic lesions (adenoma or polyps), and a control group with normal colonic mucosa. 

## 2. Materials and Methods

### 2.1. Selection of Studies and Public Data Acquisition

We searched the PubMed database with the keywords “gut microbiome AND colorectal cancer AND fecal AND metagenome NOT mice”, limited to 2019, the year when we performed the search. This generated 33 results, but after detailed reading of the manuscripts, 4 were unrelated to CRC, 10 used 16S sequencing or were related to metabolomics or viruses, and 6 were reviews or meta-analyses. Of the 13 remaining eligible studies, only eight provided access to raw data, which were selected for our analysis (Appendix A). The sequencing reads were downloaded from either European Nucleotide Archive (ENA) or Sequencing Read Archive (SRA). The studies that were included were the following: Feng et al. [17] (PRJEB7774), Vogtmann et al. [18] (PRJEB12449), Yu et al. [19] (PRJEB10878), Zeller et al. [16] (PRJEB6070), Thomas et al. [24] (SRP136711), Wirbel et al. [23] (PRJEB27928), Yachida et al. [20] (DRA006684 and DRA008156), and Gupta et al. [21] (PRJNA531273 and PRJNA397112). Small adenomas in Zeller et al., were considered healthy; stage 0 CRC in Yachida et al., were considered precancerous lesions, and normals with a prior history of colorectal surgery in Yachida et al., were excluded.

### 2.2. COLSCREEN: Study Population and Design

COLSCREEN (Colorectal Cancer Screening) study is a cross-sectional study that included 870 participants that were recruited during 2016 to 2020 from the ongoing population-based CRC Screening Program that was conducted by the Catalan Institute of Oncology. The CRC Screening Program targets men and women aged 50–69 and biannually invites them to participate using the immunochemical fecal occult blood test (FIT, OC-Sensor, Eiken Chemical Co., Tokyo, Japan). If the FIT result turns positive (≥ 20 μg Hb/g feces), the participants are referred for colonoscopy. The exclusion criteria to participate at the CRC Screening Program were gastrointestinal symptoms, family history of hereditary or familial CRC, personal history of CRC, adenomas or inflammatory bowel disease, colonoscopy in the previous five years or a FIT within the last two years, terminal disease, and severe disabling conditions. Further details on the CRC Screening Program design can be found at Peris et al. and Binefa et al. [26,27]. 

The majority of COLSCREEN participants were invited to participate after a positive FIT result, but we also invited a subset of participants with a negative FIT result (<20 μg Hb/g feces). Additionally, we recruited some patients with a clinical diagnosis of CRC in Bellvitge University Hospital (L’Hospitalet de Llobregat, Barcelona) to increase the sample size of CRC cases. Each participant provided written informed consent, underwent a colonoscopy, completed an extensive epidemiological questionnaire, and donated a blood and a fecal sample at recruitment. One week before colonoscopy preparation, the participants were asked to store a fecal sample at home at −20 °C. This sample was delivered by the participants on the day of the colonoscopy and stored at −80 °C. In the present study, we excluded those participants that reported having used antibiotics or probiotics one month before sampling. During the colonoscopy, colonic mucosa biopsy samples were obtained. The participants were classified following the criteria by Castells et al., that was used in the CRC screening programs for risk stratification as: normal, low-risk lesions, intermediate-risk lesions, high-risk lesions, or CRC [28]. For this study, a selection of 156 cases were used to define three aggregated groups: normal/no-lesions/controls (n = 51), high-risk lesions (n = 54), and CRC cases (n = 51). The ethics committee of the Bellvitge University Hospital approved the protocol of the study (PR084/16).

The fecal DNA was extracted using the NucleoSpin Soil Kit (Macherey-Nagel, Duren, Germany) following the manufacturer’s protocol. The extracted DNA quantity and quality was assessed through Qubit dsDNA Kit (Thermo Fisher Scientific, Waltham, MA, USA) and Nanodrop (Thermo Fisher Scientific, Waltham, MA, USA), respectively. The sequencing libraries were prepared with 2 µg of total DNA using the Nextera XT DN Sample Prep Kit (Illumina, San Diego, CA, USA). Sequencing was performed at 150 nucleotides, paired-end, using an Illumina HiSeq 4000 platform.

### 2.3. Bioinformatics Analysis

The human reads were removed from the metagenome samples. The raw reads were aligned to the human genome (GRCh38) using Bowtie2 [29] with options –very-sensitive-local and -k 1. Afterward, we checked the quality of the sequencing reads using FastQC (v.0.11.7) [30] and MultiQC (v.1.9) [31]. We decided to apply a first deduplication step to remove potential PCR duplicates that were present in our sequencing libraries, using clumpify (v.38.26) from BBTools [32]. Then, BBduk (v.38.26) was used to clean the reads, removing sequencing adapters, low-quality ends of reads (PHRED score < 20), and short reads after trimming (pairs where one of the reads was length < 50) were removed [32]. All the trimmed sequences were submitted again to FASTQC and MultiQC for quality control purposes (Appendix A).

The clean sequencing reads were classified using Kraken2 (v.2.1.0) [33], with a filtering threshold of 0.1, followed by Bayesian re-assignment at the species level using Bracken2 [34], with the read length parameter set at 100 or 150 depending on the study. The database files that were used for this classification correspond to those of the UHGG database v.1.0 [15].

### 2.4. Taxonomic Data Preparation

A count matrix (sample by microbial species) was created from the results of the Kraken2-Bracken2 output. This count matrix was normalized by genome length, and then the resulting microbiome dataset was subject to filtering before analysis. To do this, we transformed the count matrix to a relative abundance matrix. Then, we retained all the species that reached 0.1% abundance in each of the samples and were present in at least 5% of the samples.

Compositional downstream analyses that were performed required the replacement of zero values. For this, we used a multiplicative replacement algorithm via the *cmultRepl* function of zCompositions R package (v.1.3.4) [35].

### 2.5. Alpha and Beta Diversity

To explore the microbiome composition, we computed alfa diversity metrics measures, which were calculated using Faith’s index (*pd* function in the Picante R package) [36]. We also computed beta diversity metrics (Euclidean distances) and used multi-dimensional scaling to find general patterns in the microbiome composition. For beta diversity, we followed the steps that were suggested in Silverman et al., using the philr R package (v.1.16.0) [37]. As stated by Silverman et al., the use of Euclidean distances transformed with phylogenetic isometric log-ratio outperforms several standard distance metrics. These transformed Euclidean distances were subjected to dimensional reduction for plotting.

### 2.6. Statistical Analyses

We re-analyzed all the previously published deep sequenced stool sample metagenomes using the UHGG database [15], the Kraken2 classifier [33], and applied compositional data analysis methods to account for the compositionality of gut microbiome datasets [38].

To perform the MWAS, we used the *aldex.glm* function in ALDEx2 R package (v.1.22.0) [39], controlling for study, age, sex, and body mass index (BMI) variables, which were available across all the studies. To generate per-study estimates, we also ran ALDEx2 stratified by the study variable. The Wilcoxon rank sum test (*p*-value < 0.05) in the *aldex.ttest* function was used to identify the statistically significant CRC-enriched species, precancerous lesion-enriched species, and/or control-enriched species. We report species that showed a false discovery rate (FDR) < 0.05.

To build predictive models, the publicly available datasets were used for the discovery step, while our novel dataset was used for validation purposes. Bacterial and archaeal sequencing counts were normalized by the genome length. A cross-validated LASSO model was built using the *glmnet.cv* function in the R package glmnet (v.4.1.3) with alpha = 1. We chose LASSO because of its ability to discard unrelated variables and generate simple models. We selected the penalty parameter of the LASSO model using cross-validation. Centered log-ratio (CLR) transformed values were used as input. We trained a LASSO predictive model to assess their feasibility for predicting microorganisms that were linked to CRC, precancerous lesions, and health status. The model was trained with study, age, sex, and BMI as adjusting variables, forced into the model by setting the penalty factor to zero for those variables. We included all the species without restriction in the LASSO model because preliminary models showed that restricting to only the species FDR < 0.05 in the MWAS were suboptimal. To increase the consistency, we repeated the LASSO models 100 times with different random seeds and selected those species that were seen in at least 90 models. Our final model was defined as the average of the penalized parameters from the 100 runs. To estimate the predictive values and the area under the receiver operating characteristic curve (aROC) we only used the microbiome variables, ignoring age, sex, and BMI. If we included those variables in the model, the predictive accuracy increased, but we preferred to focus on the aROC values that were specific to the microbiome signature. The predictive accuracy of the models to discriminate between CRC cases and controls, and precancerous lesions and controls, was assessed with sensitivity, specificity, and aROC as implemented in the pROC R package [40]. The utility of the model was assessed by calculating the positive predictive values (PPV) and negative predictive values (NPV). Differences in the microbiome composition among studies were assessed using a one-way analysis of variance (ANOVA) with post hoc Tukey honestly significant difference (HSD) test (statistical significance was set at α = 5%) [41].

### 2.7. Functional Characterization

Additionally, we performed a functional analysis by aligning clean sequencing reads to the UHGP-90 database (v.1.0) using the DIAMOND aligner (v.2.0.8) [42]. For each read, the alignment with the best score was used. Then, the protein families were mapped to EGGNOG functional groups according to the classifications that were provided by UHGP, using a custom program. For simplicity, we used the functional groups ending in “@1”, which correspond to the root of the bacterial and archaeal phylogenetic trees. EGGNOG groups that did not have a “@1” mapping were discarded as they correspond to viral genetic material. The LASSO predictive functional model for CRC and for precancerous lesions included information on protein functions with a relative abundance that was higher than 5%.

## 3. Results

### 3.1. Datasets and Study Design

We identified eight studies that had analyzed the fecal microbiome using deep shotgun sequencing in patients that were diagnosed with CRC, precancerous lesions, and controls, with a minimum sample size of 20 subjects per group (Table 1). The inclusion of patients with precancerous lesions was optional, and some studies did not include this group. The studies provided information on age, sex, and BMI, which were used for further adjustment (Appendix A).

Validation of the findings was performed using 156 newly deep sequenced fecal metagenomes from our COLSCREEN study (51 CRC, 54 precancerous lesions, 51 normal controls). A summary of these data is available in Table 1 and in Appendix A, and the complete data are available in Appendix A.

### 3.2. Microbiome Description

No consistent difference in the within-sample microbial diversity was observed, neither by condition nor by study (Figure 1a). However, a geographical difference in the microbial composition was observed between the Asian and European/American samples, indicating significant differences in microbiome composition among studies (*p*-value < 0.0001) (Figure 1b) but based on the status of participants, no differences in composition were observed Appendix A.

In addition, we also inspected the alpha and beta diversity in relation to epidemiological variables (age, sex, and BMI), but these parameters were not associated (Appendix A, respectively).

### 3.3. MWAS Meta-Analysis

A total of 95 species were identified to be statistically significantly associated with CRC (FDR < 0.05, Figure 2). Most of them being control-enriched (n = 65; Figure 2a), with only 30 CRC-enriched (Figure 2b).

Among the control-enriched bacteria, we found several members of the *Faecalibacterium, Lachnospira, Blautia_A, Anaerostipes, Roseburia,* and *Coprococcus* genera. Among the CRC-associated bacteria, we found species from the *Bacteroides* genus (*Bacteroides caccae, Bacteroides nordii, Bacteroides fragilis_A,* and *Bacteroides fragilis*), *Alistipes* and *Alistipes_A* (*Alistipes senegalensis, Alistipes_A ihumii, Alistipes onderdonkii*, and *Alistipes_A indistinctus*), and other previously reported cancer-associated species such as *Parvimonas micra, Clostridium_Q symbiosium,* and *Faecalicatena torques*.

Of the 95 significant CRC-associated species, 26 were only defined in the UHGG database at the level of MAGs (20 were control-enriched and 6 were cancer-enriched). The control-enriched species belonged to the *Ruminococcaceae* and *Lachnospiraceae* families. The six CRC-enriched species that were only characterized by MAGs belonged to the *Acutalibacteraceae*, *CAG-74*, *Lachnospiraceae, Oscillospiraceae*, and *Rikenellaceae* families.

Then, we performed the same analysis on the control vs. precancerous lesion design, in the subset of four studies including data from precancerous lesions samples (231 precancerous lesions and 655 controls [16,17,20,24] (Table 1). We only observed two statistically significant species: *Lachnospira sp003537285* (*p*-value = 1.71 × 10^−05^, FDR = 0.008) and *MGYG-HGUT-00605* (*p*-value = 8.23 × 10^−05^, FDR = 0.03).

### 3.4. Predictive Models

In the predictive LASSO model, a total of 32 microbial species were consistently selected in repeated runs, of which 20 were control-enriched (negative coefficients) and 12 were CRC-enriched (positive coefficients) (Figure 3a). The statistically significant control-enriched species were *Agathobacter sp000434275, Streptococcus thermophilus*, *Blautia_A sp900066205*, *Bifidobacterium bifidum*, and *MGYG-HGUT-00213*, whereas *Dialister invisus*, *Bacteroides fragilis_A*, and *Parvimonas micra* were CRC-enriched species with *p*-value < 0.05 in the Wilcoxon rank sum test. From these 32 species, 23 of them were also identified by the MWAS meta-analysis (Appendix A). The predictive accuracy estimates that were obtained were: aROC values of 0.81 (95% CI: 0.78–0.83) for the training and 0.75 (95% CI: 0.66–0.84) for the validation in the COLSCREEN dataset (Figure 3c). This aROC value increased to 0.79 (95%CI: 0.70–0.88) when the validation model included age, sex, and BMI. At the threshold of 0.33, the specificity of the model was 0.96, but the sensitivity was 0.41. In the validation dataset, a high score was indicative of CRC presence, with a positive predictive value of 0.91, while the negative predictive value was 0.62 (Figure 3b).

The CRC-trained model was not suitable to predict the presence of precancerous lesions (aROC: 0.52, 95% CI: 0.41–0.63) (Appendix A). Therefore, we built additional LASSO models aimed at detecting precancerous lesions using two approaches: (a) including only precancerous lesion samples, and (b) including both CRC and precancerous lesion samples.

The LASSO model including only precancerous lesions finally selected 10 control-enriched and three precancerous lesion-related microbiological species obtaining aROC values of 0.71 (95% CI: 0.67–0.75) for the training step and of 0.65 (95% CI: 0.54–0.75) for the validation. The statistically significant control-enriched species were *Dorea longicatena_B*, *Lachnospira sp003537285*, and *MGYG-HGUT-01202*, while *Alistipes shahii* and *MGYG-HGUT-02726* were associated with precancerous lesions (Appendix A).

In the LASSO model including both CRC and precancerous lesions, a total of 24 species were selected, but the accuracy was not high. The aROC values were 0.76 (95% CI: 0.74–0.79) for the training step and 0.66 (95% CI: 0.56–0.75) for the validation (Appendix A).

### 3.5. Analysis of Orthologous Groups

In the protein functional analysis, we found 763 ontologies that were associated with CRC status; the vast majority (n = 655) were control-enriched. Among all the associated ontologies, the most prominent were related to translation and amino acid metabolism and transport (Figure 4a,b), while about one third (n = 250) were of unknown function.

We also built a LASSO predictive model for CRC status, using these eggNOG ontologies (Appendix A) that had an aROC of 0.79 (95% CI: 0.77–0.82) for training. When testing this model in the COLSCREEN study, the aROC was of 0.70 (95% CI: 0.56–0.81) when trying to predict CRC status (Figure 4c), and of 0.58 (95% CI: 0.47–0.69) when trying to predict precancerous lesions. In addition, we built a LASSO model that was specific for precancerous lesions and obtained an aROC value of 0.64 (95% CI: 0.59–0.68) for training. When this model was validated within the COLSCREEN study, the aROC value was 0.53 (95% CI: 0.42–0.64) when trying to predict precancerous lesions, but it scored 0.62 (95% CI: 0.52–0.73) when trying to predict CRC status (Appendix A).

## 4. Discussion

In this study, we performed an enhanced microbial classification and meta-analysis of previously published high-quality shotgun sequenced fecal metagenomes studies. We built a CRC status predictive model and validated it in our new CRC study that included 156 samples, which adds to the available datasets of similar characteristics.

We collected and re-analyzed data from eight human CRC studies with deep sequenced fecal metagenomes that were available from France, Italy, Austria, Germany, North America, China, Japan, and India. We did not include the metagenomics data from Hannigan et al., as the fecal samples were treated for virus sequencing [43]. Regarding the modeling strategy, while other studies have adopted a leave-one-study-out (LOSO) paradigm to analyze the association of the microbiome with CRC status [23,24], we opted to leave our study outside any discovery/training analysis whatsoever, and use it exclusively for validating the models that were trained in the meta-analysis.

We reanalyzed individual metagenomes from all the studies, using the UHGG database v.1.0 specific for gut microbiome [15]. Our MWAS meta-analysis identified significant associations between the 95 gut microbiome species and CRC (FDR < 0.05). Some of these species were still not completely defined in the UHGG database. The predictive accuracy of our 32 species cross-validated LASSO model was aROC 0.81 in the training meta-analysis. This predictive measure falls within the range that was reported by previously published articles (0.73–0.96) [16,17,19,20,21,22,23,24]. The aROC value was reduced to 0.75 (95% CI: 0.66–0.84) when we validated the predictive model in our completely independent COLSCREEN study but increased by 0.04 (aROC 0.79) when age, sex, and BMI were considered as additional predictors for CRC. Unfortunately, the studies meta-analyzed did not systematically provided data on other risk factors for CRC that might have led to a better predictive model. The high accuracy of our model is remarkable, since the training datasets included different populations with diverse ethnicity, environmental, and lifestyle exposures, and the novel testing dataset was geographically independent of the training ones. Additionally, our model showed a high specificity (0.96) at a threshold of 0.33, suggesting that scores that are higher than this value are indicative of the presence of CRC. However, the model had low sensitivity for CRC and was no better than chance for precancerous lesions (aROC: 0.52, 95%CI: 0.41–0.63).

In the case of the predictive model that was specific for precancerous lesions, the accuracy measures fall within the range of the previous published similar predictive models [17,24]. Although our study was cross-sectional, and we cannot establish the direction of causality, the fact that our model was only valid for CRC and not for precancerous lesions perhaps indicates reverse causality regarding the association of our microbiome signature and CRC (i.e., CRC is causing the shift in the observed microbiome composition, and not the other way around). It is also possible that the identified microbiome signature plays a more important role in CRC progression, rather than initiation. This would imply a higher potential utility in prognosis and as a target for CRC-associated treatment, rather than for CRC detection. However, it should also be noted that this interpretation is limited to the signature that was identified by our model, and other sources of evidence suggest a causal role of the microbiome in CRC. For example, one experimental study showed that human feces gavage from CRC patients can induce dysplasia in mice [44]. Another limitation regarding causality is that our study focused on the microbiome that was analyzed in stool, which may differ from that in the mucosa which may play a more direct role. Also, less data were available for precancerous lesions than for CRC, which reduced the statistical power to identify a good signature for them.

Other metagenomics analyses of CRC datasets have attempted to identify microbial diagnostic signatures. Despite following different methodological strategies, most of them have obtained comparable results. For instance, *Fusobacterium nucleatum* [16,18,19,20,24], *Peptostreptococcus stomatis* [16,17,20,21,24], *Gemella morbillorum* [17,20,23,24], *Peptostreptococcus anaerobius* [19,20,24], *Solobacterium moorei* [23,24,24], *Prevotella intermedia* [23,24], *Parvimonas micra* [17,19,20,21,23,24], and *Bacteroides fragilis* [16,21,24] have been formerly identified in almost all studies. Our results support the fact that the latest two species are extensively associated with CRC across all cohorts and should be considered for future microbiome-based CRC diagnosis development, as suggested by others [20,23,24,45].

Furthermore, our study classified the species *Sutterella wadsworthensis_A, Anaerotignum sp000436415,* and *Dialister invisus* as CRC-associated bacteria, contributing to the evidence of their pathogenic role [46,47,48]. It should be noted that the genus *Alistipes* was consistently classified as disease-enriched across all the precancerous lesion models [49].

Besides, *Streptococcus thermophilus* and *Bifidobacterium bifidum* among others, were selected by the LASSO predictive model as control-enriched species, in agreement with previous results [17,50]. The role of control-enriched species in CRC development is still diffuse. However, if confirmed by further studies, they could be candidates for cancer prevention strategies. For example, *Faecalibacterium prausnitzii*, a known butyrate producer [51], or *Streptococcus thermophilus,* that is a folate producer and has been associated, in conjunction with *Bifidobacterium bifidum,* with positive effects on gastrointestinal disorders [52,53]. Interestingly, both *Streptococcus thermophilus* and *Bifidobacterium bifidum* together with *MGYG-HGUT-00605* and *Lachnospira sp003537285* were selected as control-enriched species in all models that we performed.

The analysis of functional data (i.e., information about gene families or orthologous groups that were observed in microbiome samples) remains challenging, in part due to the vast number of categories that were considered in these analyses, many of which still are classified as of unknown function. There is still research that is needed for improving our knowledge of the genic functions that are present in the gut microbiome [10]. Regardless of this limitation, the presence of certain orthologous groups could be used for CRC prediction, as shown by our predictive model with an aROC value of 0.70 but not for precancerous lesions (aROC: 0.58). The three main categories, based on the most curated eggNOG ontologies, were: (1) translation, (2) amino acid metabolism and transport, and (3) cell wall/membrane/envelope biogenesis. Recently, *Casimiro-Soriguer* et al., observed that membrane proteins were the most relevant eggNOG features [45]. Other authors have also attempted to establish functional signatures for CRC discrimination, predominantly using the Kyoto Encyclopedia Genes and Genomes (KEGG) database. Despite this dissimilarity and the wide range of ROCs values (0.70 and 0.96), the amino acid metabolism and transport pathways were repetitively identified across studies [16,18,19,20,21,22,23,24]. The predictive accuracy of a specific non-colonic lesions model was also insufficient (aROC 0.58).

For this study, we decided to use our enhanced bioinformatics pipeline for gut microbiome classification. We used the Kraken2 classifier algorithm, which queries each sequencing read against a database of full genomes using k-mers. On the other hand, while it is common to use RefSeq as a database, we chose the UHGG database (v.1.0) which offers the advantage of including MAGs, making it more systematic for gut microbiome analysis. In this study, we found certain species that were still undefined in UHGG to be statistically associated with CRC, which demonstrates that yet uncharacterized microbes might harbor bacteria with clinical interest [10]. Other advantages of this database are the use of the genome taxonomy database (GTDB) taxonomic annotations, which better reflect evolutionary relationships, and the fact that it is derived from fecal samples, reducing the risk of misclassifications.

## 5. Conclusions

Our most important contribution is a signature of 32 microbial species that has good predictive accuracy to identify CRC status. The signature is robust, and we validated it in a new well-characterized independent dataset. However, its applicability in CRC screening programs is still doubtful, as the sensitivity is low, and its capacity to identify precancerous lesions is less accurate. This suggests that the microbial species that is enriched or depleted in CRC of this signature are merely a consequence of the tumor and not the initial gut epithelial malignant transformations. Thus, further research in defining microbial signatures that considers both CRC and precancerous lesions is warranted.

## Figures and Tables

**Figure 1 cancers-14-04214-f001:**
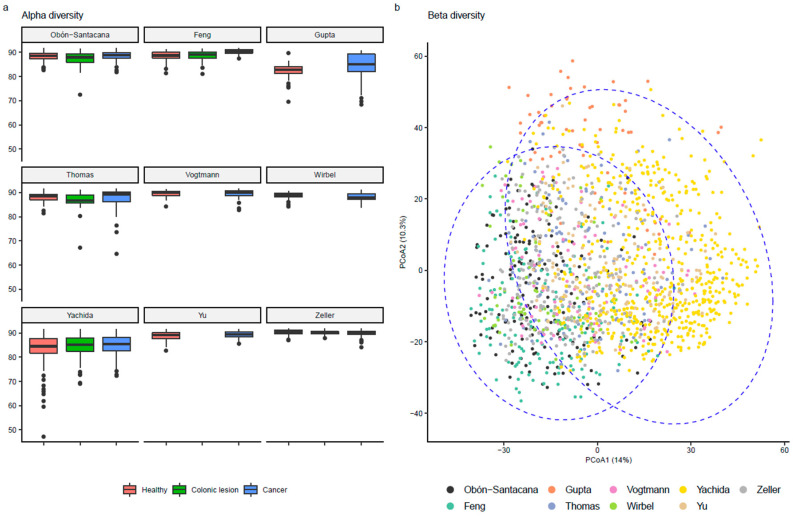
Microbiome diversity statistics of the included metagenomic datasets. (**a**) Alpha diversity metrics (Faith’s index). (**b**) Beta diversity metrics (based on Euclidean distances of ILR-transformed relative abundance counts). The right ellipse represents Asian studies (Gupta, Yachida and Yu) meanwhile the left ellipse depicts USA and EU studies (Feng, Obón-Santacana, Thomas, Vogtmann, Wirbel and Zeller). Both ellipses represent a 95% confidence region.

**Figure 2 cancers-14-04214-f002:**
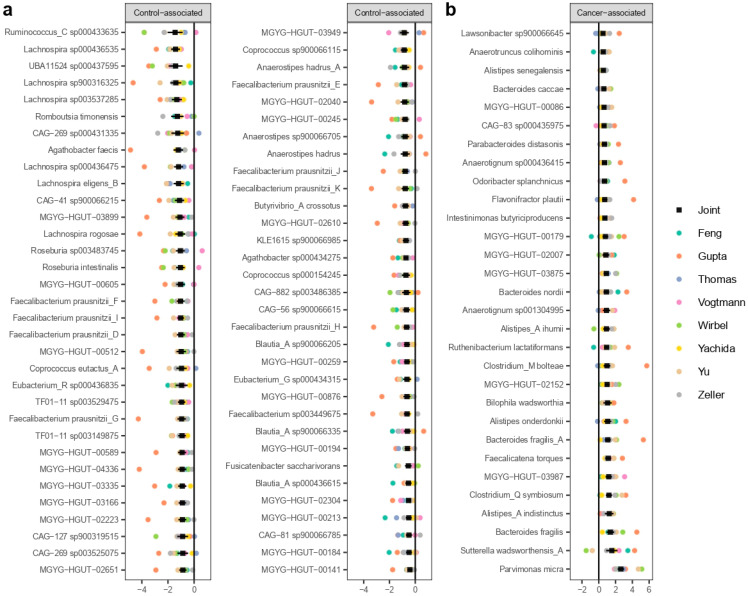
Species that were statistically associated to CRC. Black squares with lines represents the estimate of the effect size and 95% confidence intervals. The colored dots represent the estimates of the effect sizes for each dataset. (**a**) Species that were found to be decreased in cancer (columns 1 and 2). (**b**) Species that were found to be increased in cancer (column 3).

**Figure 3 cancers-14-04214-f003:**
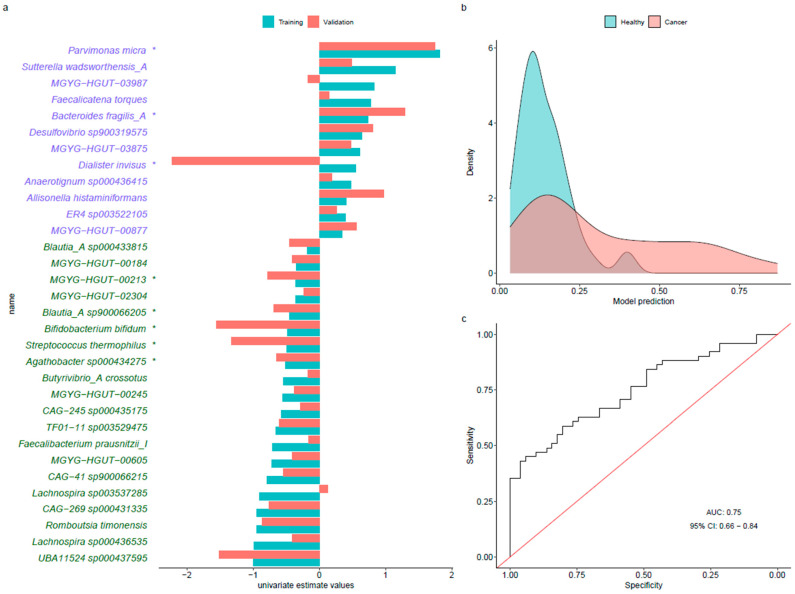
Summary of the LASSO predictive model in our dataset. (**a**) Training (blue) and validation (red) estimate values for each control-enriched (green) and cancer-enriched (purple) species selected by the model. (**b**) Density plot of model prediction, colored by the status of the samples. (**c**) Receiver operating characteristic curve representing the performance of the model. *: statistically significant (*p*-value < 0.05) based on Wilcoxon rank sum test in *aldex.ttest*.

**Figure 4 cancers-14-04214-f004:**
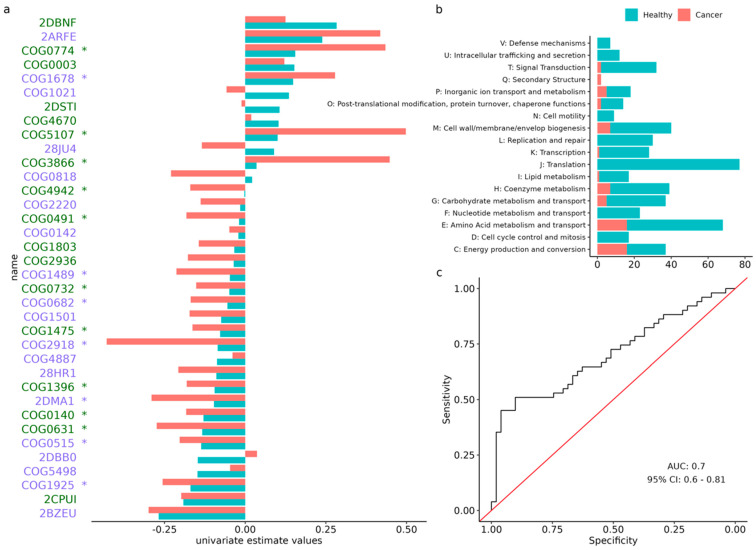
Analysis of eggNOG orthologous group. (**a**) Training (blue) and validation (red) estimate values for each control-enriched (green) and cancer-enriched (purple) species that was selected by the model. (**b**) Amount of significantly associated orthologous groups, clustered by general category. Blue represents the control-associated groups, while red represents the cancer-associated groups. Category “S” (function unknown) was excluded. Orthologous groups belonging to more than one category were counted for each. (**c**) The receiver operating characteristic curve representing the performance of the predictive model. *: statistically significant (*p*-value < 0.05) based on Wilcoxon rank sum test in *aldex.ttest*.

**Table 1 cancers-14-04214-t001:** Summary of sample sizes and epidemiological data of all the included studies.

	Ref	Total	Healthy/ Negative	Precancerous Lesions	CRC Cases	Woman	Age	BMI
Study		n	n	n	n	%	Mean (SD)	Mean (SD)
Zeller et al.	[16]	199	93	17	89	41	62.3 (12.1)	25.6 (4.0)
Feng et al.	[17]	156	63	47	46	44	66.9 (8.3)	27.4 (4.0)
Vogtmann et al.	[18]	104	52	-	52	29	61.5 (12.3)	25.1 (4.2)
Yu et al.	[19]	128	54	-	74	37	64.2 (9.1)	23.8 (3.1)
Yachida et al.	[20]	576	251	140	185	40	61.9 (11.0)	22.9 (3.4)
Wirbel et al.	[23]	82	60	-	22	48	60.0 (11.6)	25.0 (3.7)
Thomas et al.	[24]	140	52	27	61	35	63.5 (9.7)	25.6 (4.0)
Gupta et al.	[21]	59	30	-	29	51	50.8 (16.1)	21.5 (3.1)
Obón-Santacana et al.	-	156	51	54	51	36	61.0 (7.9)	27.6 (4.2)

## Data Availability

The dataset that was generated and analyzed in our study is available at the Zenodo repository (https://doi.org/10.5281/zenodo.6671562, accessed on 27 August 2022). The raw data are available on reasonable request from the corresponding author (VM, email: v.moreno@iconcologia.net).

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
