# Peer review of "Meta-Analysis and Validation of a Colorectal Cancer Risk Prediction Model Using Deep Sequenced Fecal Metagenomes"

_cancers, 2022, doi:10.3390/cancers14174214_

Round 1

Reviewer 1 Report

Herein, the authors have systematically built a model using all available shotgun metagenomics CRC datasets and tested its predictive value using the COLSCREEN dataset. This study is interesting and relevant and the conclusions drawn are accurate and appropriate. 

Aside from thorough editing for English language, I have only one critique that should be addressed in the text if not explored in supplementary materials. In lines 201-205, the authors state that they ignore age, gender, and BMI despite those variables improving the model. Taking this approach insinuates that microbiota are the only causative factor in CRC and biological variables do not contribute to risk. Thus, these confounding variables should be included (at least supplementally) to estimate predictive values and aROC. It would be beneficial to also explore whether COLSCREEN data are predicted by the metagenomes from corresponding regions only.

Author Response

We have extensively revised the manuscript to improve English style. We also have unified the term "precancerous lesions" that previously had multiple varieties (adenoma, non-CRC, premalignant, preneoplastic).

Regarding the inclusion of covariates other than the microbiome in the predictor model, we already had reported in results (lines 289-90) “The aROC value increased to 0.79 (95%CI: 0.70-0.88) when the validation included age, sex, and BMI (data not shown).” We now include the coefficients for these variables in supplementary table 3, in case a researcher wants to use them in the predictor. Age, sex and BMI were the variables that we could systematically obtain from the studies meta-analyzed. Obviously, other risk factors for colorectal cancer like alcohol, smoking, meat and vegetable consumption, physical activity, regular use of aspirin and family history could have improved the predictive accuracy of the model, as we have shown in other papers. Unfortunately, these variables were not available in the training datasets. We have added a sentence to the discussion to stress this.

Reviewer 2 Report

The current study performed a meta-analysis of shotgun sequenced fecal metagenomes and conducted an external validation in a CRC cohort. They are to be recommended for the study design and robust methodology. Certain minor issues need to be addressed.

1. Searching strategy should be more specific. The authors are advised to add searching alogrithm presented with AND, OR. A simple flow chart demonstrating the inclusion and exclusion procedure could also tell the readers how was the 8 studies selected to conduct the analysis.

2. A signature of 32 bacterial species that have a good 32 predictive accuracy to identify CRC but not preneoplastic lesions. The authors conferred the enriched or depleted species were consequences of CRC. This statement should be contemplated as the current study did not evaluate the mucosal microbiome. Moreover, previous study have shown human CRC feces gavage could result in mice dysplasia.

Author Response

  1. We have included in the methods section a detailed description of the strategy used to identify the studies included in the meta-analysis. Also, a PRISMA chart has been added as supplementary material (S1 Figure)
  2. We agree with this reviewer that the conclusion that the microbiome signature is a consequence and not a cause of cancer is indirect. Our study is cross-sectional and cannot infer the direction of causality. The interpretation derives from the fact that if the signature was causal, it should also determine precancerous lesions formation and, thus, should have some predictive value for precancerous lesions. We have revised the discussion to better explain that our interpretation if for the identified signature, not for the microbiome in general, and include the suggested limitations.